# Second-Order Scattering Quenching in Fluorescence Spectra of Natural Humates as a Tracer of Formation Stable Supramolecular System for the Delivery of Poorly Soluble Antiviral Drugs on the Example of Mangiferin and Favipiravir

**DOI:** 10.3390/pharmaceutics14040767

**Published:** 2022-03-31

**Authors:** Mariya A. Morozova, Vladimir N. Tumasov, Ilaha V. Kazimova, Tatiana V. Maksimova, Elena V. Uspenskaya, Anton V. Syroeshkin

**Affiliations:** Department of Pharmaceutical and Toxicological Chemistry, Medical Institute, Peoples’ Friendship University of Russia (RUDN University), 6 Miklukho-Maklaya Street, 117198 Moscow, Russia; vyldemar@mail.ru (V.N.T.); kazymova-iv@rudn.ru (I.V.K.); maximova-tv@rudn.ru (T.V.M.); uspenskaya-ev@rudn.ru (E.V.U.); syroeshkin-av@rudn.ru (A.V.S.)

**Keywords:** humic substance, drug delivery system, nanoparticles, antiviral drugs, mangiferin, favipiravir, second-order scattering quenching

## Abstract

In the present work, the methods of dynamic light scattering and fluorescence spectroscopy were applied to study the optical properties of aqueous dilutions of the humic substances complex (HC) as a potential drug delivery system. The supramolecular structures in the humate solution were characterized as monodisperse systems of the submicron range with a tendency to decrease in particle size with a decrease in the dry matter concentration. The slightly alkaline medium (8.3) of the studied aqueous dilutions of HC causes the absence of a pronounced fluorescence maximum in the region from 400 to 500 nm. However, the presence of an analytically significant, inversely proportional to the concentration second-order scattering (SOS) signal at 2λ_ex_ = λ_em_ was shown. In the examples of the antiviral substances mangiferin and favipiravir, it was shown that the use of the humic complex as a drug carrier makes it possible to increase the solubility by several times and simultaneously obtain a system with a smaller particle size of the dispersed phase. It has been shown that HC can interact with mangiferin and favipiravir to form stable structures, which lead to a significant decrease in SOS intensities on HC SOS spectra. The scattering wavelengths, λ_ex_/λ_em_, were registered at 350 nm/750 nm for mangiferin and 365 nm/730 nm for favipiravir, respectively. The increments of the scattering intensities (I0/I) turned out to be proportional to the concentration of antiviral components in a certain range of concentrations.

## 1. Introduction

Targeted drug delivery is far from being a new research and production task for chemists and pharmacologists, but it still seems to be very relevant; the design and synthesis of efficient drug delivery systems are of vital importance for medicine and healthcare [1]. Generally, carrier molecules are characterized by nanometer linear dimensions, which allow them to overcome the limitations of free therapeutics and navigate biological barriers on different levels [2]. Nanosystems are represented by varied morphologies: nanocapsules, nanospheres, liposomes, dendrimers, hydrogels, etc., but regardless of structure, they all are aimed to decrease the toxicity of therapeutics and improve their delivery by overcoming poor bioavailability and low tissue absorption [3,4,5,6]. The choice of the delivery system substantially determines the properties of the finished product. Compared with artificial materials, natural materials exhibit superiority in biocompatibility, easy accessibility, and modification, often possessing low toxicity and potentially favorable pharmacokinetics [7,8]. Nonetheless, the tremendous potential of natural polymers as drug carriers is still under-represented and deserves more attention [1].

Among natural polymers, one can distinguish macromolecules of humic acids (HAs), that comprise humic substances, which are widely distributed in terrestrial soil, natural water, and sediments, as a result of the decay of natural residues [9]. Although the complex and heterogeneous nature of HAs prevents an accurate characterization of their chemical composition, it is accepted that these carbon-containing compounds are flexible linear polymers with carboxyl and phenol as the major functional groups, so at neutral pH, HAs are known to form clusters stabilized by the hydrophobic effect and intermolecular hydrogen bonds [10]. It is believed that peripheral pores in HA polymers can accommodate natural and synthetic organic chemicals in a clathrate-type arrangement. According to [11], biomaterials such as Has extend solubility, permeability, and dissolution in various drug combinations. For instance, in [12], the bioavailability of carbamazepine was improved by complex formation with humic substances, ketoconazole’s dissolution rate was significantly enhanced in a complex with humic acid due to its micellization nature [13], the interaction between humic acid salts and papaverine hydrochloride, benzohexonium and B-group vitamins was experimentally discovered in [14], etc.

At the same time, HAs cannot be considered simply an inert carrier of an active drug; these compounds are exceptional in the diversity of their pharmacological activity and can often have a potentiating effect on the delivered therapeutics [15]. HA activity against both naked and enveloped DNA viruses was already proved in preliminary in vitro studies with Coxsackie A9 virus, Influenza A virus, Herpes simplex virus type 1, immunodeficiency virus type 1 and type 2, and cytomegalovirus [11]. The recently established specific virucidal activity of HAs against a new severe acute respiratory syndrome, coronavirus 2 (SARS-CoV-2) [16], makes these natural polymers of particular interest, as HAs could simultaneously act as a transport system for antiviral drugs and at the same time enhance their therapeutic effect due to their own active properties. This is especially important given the fact that most antiviral drugs currently on the market are oral tablets, albeit their active ingredients have poor oral bioavailability due to low solubility/permeability [17]. Therefore, loading antiviral drugs into advanced delivery systems, such as HAs, may be conducive to overcoming the abovementioned disadvantages.

This paper presents some new results describing the process of interaction between the complex of HAs and two antiviral pharmaceuticals with different molecular structures. The first one, favipiravir (5-fluoro-2-oxo-1H-pyrazine-3-carboxamide), is a synthetic drug and a nucleoside analog, possessing the ability to bind and inhibit RNA-dependent RNA polymerase, which ultimately prevents viral transcription and replication. Favipiravir demonstrated good recovery in some of the cases of COVID-19 [18], but its full potential as a therapy for coronavirus remains to be determined, including the optimal administration timing, dosage, and duration of therapy [19,20,21]. The second, mangiferin (1,3,6,7-tetrahydroxy-C2-β-D-glucoside), is a xanthone glucoside found at a significant level in higher plants of various genera; it possesses immune-modulating effects on different oxidative mechanisms in various disorders [22]. In the Russian Federation, the substance is officially registered (number of state registration-LSR-008603/09-060820) under the name Alpizarin^®^ as a drug with antiviral activity against Herpes simplex type 1 and 2, Herpes zoster, Varicella zoster, and cytomegaloviruses. As for coronavirus therapy, mangiferin was one of those phenolic compounds that have shown in silico binding affinity with angiotensin-converting enzyme 2, a potent molecular receptor for SARS-CoV-2 [23,24]. Additionally, mangiferin is found to provide a promising therapeutic effect on treating and managing respiratory inflammation, as it protects against lung injury and inflammatory response by inhibiting NLRP3 inflammasome activation in macrophages [25,26].

These drug substances are close in that the delivery of both is challenging because of their limited solubility, and their formulation is difficult with common organic solvents and water [27]. Mangiferin, following the biopharmaceutical classification, can be classified as a class IV compound with low solubility and low bioavailability [28]. Certain solubilization technologies such as nanoemulsions, phospholipid complex, cyclodextrin inclusion complex, and liposome can improve the dissolution and bioavailability of mangiferin, but a large number of excipients might lead to potential toxicity problems [29]. Favipiravir also has poor aqueous solubility, but according to the data obtained from the in vitro studies, it can be considered as a representative of class I compounds [17,30]. On oral administration, it is absorbed well, having bioavailability of more than 90%; however, high doses of orally administered tablets (3600 mg/day) can increase the risk of adverse effects [27,31]. Therefore, a simple and effective approach needs to be developed to create a versatile low-toxic delivery system of favipiravir or mangiferin. For those purposes, we propose using natural macromolecules of HAs as they provide a modification of the environment for guest molecules, changing their properties.

## 2. Materials and Methods

### 2.1. Reagents

#### 2.1.1. HAs

We used a liquid concentrated complex of humic–fulvic acids isolated from lowland peat according to the patented technology of the company VimaVita (LLC System-BioTechnologies, Moscow, Russia). Concentrated humic complex (HC) containing purified water and active components—HAs, hymatomelanic acids, fulvic acids, and structural analogs of humic substances, was obtained by oxidative-hydrolytic degradation of lignin-containing raw materials followed by high-intensity acoustic cleaning. The test preparation was a concentrated dark brown viscous liquid with pH = 7.98 ± 0.1 and dry matter content 7.34–10^−2^ g/mL [16]. To study the properties of the humic complex as a delivery system for antiviral drugs, the original preparation was not used, but its aqueous dilutions in the ratio from 1:500 to 1:3000 by volume. For dilution, highly purified water was used, obtained using the Milli-Q^®^ purification system (Merck, Darmstadt, Germany). All investigated solutions were stored at room temperature for no longer than 24 h.

#### 2.1.2. Bovine Serum Albumin (BSA)

Small, stable, and moderately inert protein BSA was employed as a colloidal dispersion reference (CAS 9048-46-8, molecular weight 66 kDa, Santa Cruz Biotechnology, Inc., Dallas, TX, USA). Samples for studying fluorescence quenching and evaluating the dispersion composition depending on the concentration of the solution were prepared by dissolving dry BSA in water (Milli-Q^®^ purification system) starting from a maximum concentration of 10 mg/mL.

#### 2.1.3. Pharmaceutical Substances

The functionality of the delivery system based on the HC was determined using two antiviral substances—favipiravir (LLC “CHROMIS”, Moscow, Russia), and mangiferin produced by Pharmcenter VILAR CJSC (Moscow, Russia). Both substances passed all the control stages and were of the proper quality. Solution concentration for the tested substances was selected based on the solubility values presented in the regulatory documentation: slightly soluble (100–1000 mL/g) and practically insoluble (more than 10,000 mL/g) in water favipiravir and mangiferin, respectively.

### 2.2. Fluorescence and Scattering Spectroscopy

The method of fluorescence spectroscopy was applied for a comparative study of the properties of colloidal systems of albumin and humic complex during dilution, as well as for studying the phenomenon of second-order scattering (SOS) quenching during the interaction of active substances with a delivery system based on humates. Fluorescence measurements were performed using a 1 cm quartz cell on a Cary Eclipse spectrofluorometer (Agilent Technologies, Inc., Santa Clara, CA, USA) with two ultrafast scanning monochromators. The width of the excitation and emission slit was adjusted at 5 nm. Exciting wavelengths were selected considering the literature data and the results of the analysis of the test compounds by the method of absorption spectroscopy (Cary 60, Agilent Technologies, Inc., USA) [32,33]. The excitation and emission/scattering wavelengths were, respectively, for HC $\frac{\lambda_{\mathrm{ex}}}{\lambda_{\mathrm{em}/\mathrm{sos}}}=\frac{350\;\mathrm{nm}}{460/700\;\mathrm{nm}}$, BSA $\frac{\lambda_{\mathrm{ex}}}{\lambda_{\mathrm{em}/\mathrm{sos}}}=\frac{280\;\mathrm{nm}}{340/560\;\mathrm{nm}}$, favipiravir $\frac{\lambda_{\mathrm{ex}}}{\lambda_{\mathrm{sos}}}=\frac{365\;\mathrm{nm}}{730\;\mathrm{nm}}$, and mangiferin $\frac{\lambda_{\mathrm{ex}}}{\lambda_{\mathrm{sos}}}=\frac{350\;\mathrm{nm}}{700\;\mathrm{nm}}$.

### 2.3. Dynamic Light Scattering (DLS)

A Zetasizer Nano ZSP (Malvern Panalytical, Worcestershire, UK) based on dynamic light scattering was used to measure the size of nanoparticles in the colloidal systems of albumin, humic complex, and HC with dissolved antiviral drugs. For this purpose, aqueous dispersions of colloidal polyelectrolyte nanoparticles with concentrations from 1.47 × 10^−4^ to 2.45 × 10^−5^ g/mL for HC and 1 × 10^−2^ to 1 × 10^−5^ g/mL for BSA, respectively, were prepared. Solutions of the tested substances in HC at maximum concentrations, which it turned out to be possible to obtain in the experiment (1 × 10^−4^ g/mL for mangiferin and 2 × 10^−3^ g/mL for favipiravir), were also examined for the presence of nanostructures. Disposable polystyrene cuvettes, filled with 1 mL of sample, were used. For each size determination, three replicate measurements were performed, and the average size value was calculated. Each measurement consisted of 12 runs. The refractive index value was 1.334.

### 2.4. Data Processing

Data analysis was acquired from *n* ≥ 3 independent experiments and is presented as the mean ± standard deviation (SD). Data processing and plotting was performed using OriginPro 2017 (OriginLab, Northampton, MA, USA) software.

## 3. Results and Discussion

### 3.1. Dispersion Analysis of the Supramolecular Structure of Aqueous Dilutions of the HC by the DLS Method

The idea of this work was to show on the example of a liquid humic complex that the formation of a stable system with dissolved drugs is possible even in humic dilute solutions. The technology should provide certain advantages in drug delivery, compared to aqueous solutions of the same substances, especially in issues of particle size optimization and concentration booster. To do this, the initial concentration of the HC was diluted with water under the manufacturer’s instructions to obtain a solution ready for oral administration. When choosing a suitable dilution for the experiment, we based it on the physicochemical and virucidal properties of the HC established earlier [16], and additionally considered the dimensional characteristics of the finished disperse system, which changed depending on the dilution. So, using the method of dynamic light scattering (photon correlation spectroscopy), widely used to determine the size of macromolecules moving randomly in a medium [34,35], it was shown that the diameter of the particles of the humic complex decreases with dilution (Figure 1).

This result confirms contemporary concepts of molecular and supramolecular structure of humic substances: they are not macromolecular polymers but rather superstructures of apparent large size and self-assembled by relatively small heterogeneous molecules held together by mainly hydrophobic dispersive forces [36,37].

Another compatible theory describes a surfactant-like aggregation of humic molecules by forming micelles with interior hydrophobic regions and exterior hydrophilic regions of amphiphilic humic molecules [38]. Because of the amphiphilic nature, humates tend to organize spontaneously in an aqueous solution, forming micelle-like structures akin to those formed by synthetic surfactants. The humic micelles do not, however, consist of identical monomers, but rather of a variety of species of different molecular sizes [39].

It follows from Figure 1 that when the humic complex is diluted, even within one order, the supramolecular structure degrades with the release of smaller particles. Thus, a decrease in concentration from 1.47 × 10^−4^ (1:500) to 2.45 × 10^−5^ (1:3000) g/mL leads to a decrease in the diameter of existing particles by 150 nm. Dilution of HC in a volume ratio of 1 to 2000 (3.67 × 10^−4^ g/mL) corresponds to the smallest particle size (d = 296 ± 23 nm) while maintaining a monomodal distribution; subsequent dilutions (1:2500, 1:3000) give a polydisperse system with two pronounced size maxima.

Monodispersity while maintaining a tendency to decrease in particle size is a clear advantage of the humic complex considered as a drug delivery system. So, for example, proteins, as well as humic substances, but due to the presence in their structure of ionized residues of amino acid molecules, exhibit certain properties common to aqueous polyelectrolyte solutions [40]. Consequently, they also can emerge as potential drug delivery systems, overcoming various limitations of conventional therapy [41,42]. However, when researching the behavior of a standard colloidal solution of BSA in an aquatic environment, we were questioned regarding the dispersion stability and particle size distribution of the protein solution [43].

BSA, like many common globular proteins, is very small in hydrodynamic size, with a monomer diameter of nearly 7 nm [44]. Dynamic light scattering data confirmed the presence of particles less than 10 nm in size in the initial protein solution, but we also identified the presence of aggregates in the protein solution appearing after dilution by two orders (Figure 2).

The initial solution of albumin (1 × 10^−2^ g/mL) has a practically neutral pH value; however, when the solution is diluted in order, acidification of the medium is observed due to insufficient buffer capacity of the protein. As we approach the isoelectric point (IEPBSA = 4.5–5.0), the negative charge of the protein molecule is gradually neutralized, which leads to the loss of the aggregative stability of the system [45]. As shown in Figure 2, at pH 7.0, in-stock solution, hydrated BSA molecules do not aggregate due to electrostatic repulsions and have a size distribution of 2 and 24 nm, which is comparable to the size and shape of individual BSA molecules (14 nm × 4 nm × 4 nm) [46]. At pH 5.40, at maximum dilution, DLS reveals the BSA molecules aggregate and show multiple size distributions.

When diluting the humic complex in the range from 1:100 to 1:10,000, no changes in the pH of the solution occur—a stable value of 8.33 remains (Figure 1). So, unlike BSA, the structural features of humic substances make it possible to stabilize the pH value during dilution and form a monodisperse system: dilutions of HC demonstrate a shift of the maximum distribution on the curve I, %—d, nm to the left while an increase in the absolute value of the negative ζ-potential shows stability maintenance [16]. This may be due to an increase in the thickness of the double electric layer because of a decrease in the concentration of counterions in the diffusion layer of polyelectrolyte solutions [16]. The results obtained make it possible to substantiate the advantages of humic substances from the point of view of their use as a drug delivery system.

### 3.2. Fluorescence and Scattering Quenching in Solutions of HC Supramolecular Structures

Spectral methods are a powerful tool to reveal the binding affinities of drugs at low concentrations. In particular, the fluorescence quenching technique is considered a method for measuring binding affinities and monitoring the molecular interactions because of its high sensitivity, reproducibility, and relative ease of use [47]. Thus, concentration fluorescence quenching is used in the study of quantum dots, protein interactions, formation of a ligand–receptor bond, formation of nanoparticles, liposomes, etc. [48,49,50]. However, to apply this approach to the studied transport system based on the humic complex, it is necessary to characterize the fluorescent properties of the solvent itself and its aqueous dilutions, as well as pure water used. It is known that humic substances, as compounds with a mostly polyphenolic structure, are capable of fluorescing upon excitation in the range from 380 to 480 nm [51,52,53]. The intensity of the emission band in the spectrum in the case of humates depends on the pH of the medium: most phenols fluoresce in a neutral or acidic medium, while in an alkaline medium, deprotonation of phenolic hydroxyls leads to the formation of nonfluorescent phenolate ions. Thus, in previous work, we observed this phenomenon during an experiment with fulvic acids, aqueous dilutions of which had a stable pH value below 6 units, while a pronounced intense maximum was observed in the fluorescence spectrum in the region of 460 nm [16]. In the case of HC water dilutions, the working pH range is more than 8 units. As can be seen from Figure 3, the standard emission band of humic compounds on the fluorescence spectra of HC aqueous dilutions (1:500–1:3000) is weakly manifested, while it has an inverse concentration dependence.

Typically, in dilute solutions, the fluorescence intensity is directly proportional to the fluorophore concentration, but here we have observed the aggregation-caused quenching that confirms the results obtained by the method of dynamic light scattering—dilution of the initial preparation of the humic complex leads to an increase in the number of fluorophore nanoparticles. However, the observed dependence is not linear, the maximum fluorescence value is achieved when the initial preparation is diluted by 1000 times, which corresponds to a dry matter concentration of 2.94 × 10^−5^ g/mL (Figure 4, red line); subsequent dilutions by 2–3 times do not lead to signal growth. We have shown that at very low HC concentrations (10^−14^ g/mL) the fluorescence band completely disappears and even a shift of the excitation radiation length to a shorter wavelength region (280 nm) does not lead to the appearance of a fluorescence signal in the spectrum of humates (Figure 4, inset). This result can be explained both by the low content of fluorophores in the solution and, probably, by the effect of the pH.

On the emission spectra of HC dilutions, another intense maximum was found for which 2λ_ex_ = λ_em_. This signal is not related to fluorescence, but rather refers to one of the types of elastic light scattering, which the technical capabilities of modern fluorometers allow the registering of second-order scattering [54,55]. The second-order scattering presenting at the double wavelength of the excitation wavelength is a common phenomenon and it has been always regarded as a kind of interference of spectrofluorometry while the characteristics and principles of SOS had not been studied deeply [56]. In the last decade, the analytical techniques of resonance Rayleigh scattering, second-order scattering, and frequency doubling scattering have been given much attention because of their high sensitivity, simplicity, and rapidness. Similar to fluorescence, scattering methods have been applied to the study of macromolecules, nanoparticles, quantum dots, determination of inclusion constant, etc. [57]. Considering all the above arguments, as well as the fact that humates are supramolecular structures that form nanoparticles upon dilution, we decided to analyze the obtained SOS data in more detail.

A real connection between the discussed maximum and scattering was shown, since when the wavelength of the exciting radiation was changed from 350 to 280 nm, the second-order scattering maximum synchronously shifted from 700 to 560 nm, i.e., the ratio 2_λex_ = λ_em_ remained constant, which is not typical for fluorescence. We concluded that the obtained SOS signal depends strictly on the content of humates in the solution, reaching the limit values at very high dilutions, which is probably due to an increase in the number of scattering nanoparticles (Figure 4, black line). It is important to point out that while studying the fluorescence spectrum of pure water at excitation wavelengths from 250 to 365 nm, we did not find any scattering maxima at 2λ_ex_ = λ_em_ exceeding 1 a.u. (Figure 5). Thus, it was shown that humic substances that weakly fluoresce in the UV region have a high-intensity second-order scattering signal, which depends almost linearly on concentration in the inverse order.

The result obtained for the humic complex again, as in the case of dispersion analysis, did not coincide with the results for the traditional colloidal system based on BSA. The insert in Figure 6 shows the normalized intrinsic BSA emission at concentration levels from 10^–5^–10^−2^ g/mL. Tryptophan emission dominates BSA fluorescence spectra in the UV region with λ_max_ near 340 nm. Its fluorescence increases linearly with BSA concentration up to 10^−3^ g/mL and starts decreasing at higher concentrations (Figure 6, red line). Most likely, we observe the process of the concentration quenching of fluorescence with an increase in tryptophan fluorophore amount [58]. It is a well-known fact that scattering is many times more common than fluorescence; however, if the molecule fluorescence is tremendously high it is almost impossible to observe the scattering signals. On the other hand, in the weakly fluorescent molecules, the scattering signals are comparable to, and sometimes even higher than, the fluorescence [59]. So, the tryptophan fluorescence expressed in the BSA molecule is quite intense but still allows us to see the second-order scattering signal at 500 nm. Its value also depends on the protein concentration, however, not as in the case of humates, but strictly repeating the behavior of the fluorescence signal—first, growth with increasing concentration, then quenching (Figure 6).

The fact that the study of tryptophan fluorescence of proteins is widely used to assess the structural state of molecules is obvious, but methods based on measuring the SOS of proteins also successfully exist [56]. That is why it is quite likely that the signal from scattering on the fluorescence spectra of the humic complex, which is much more pronounced than the fluorescence itself, should carry significant information about the processes in the system and can be used to assess the binding affinity of HC. Completing the stage of the spectral and dimensional characterization of aqueous dilutions of the HC, we selected a dilution 1 to 2000 for further work with antiviral drugs, because it contains particles of the smallest size in a monomodal distribution (Figure 1), and the concentration-dependent SOS signal is significant enough to work with (Figure 3).

### 3.3. Complexation of Liquid HC (Dilution 1:2000) with Antiviral Drugs

The ability of the humic complex to exhibit the properties of the delivery system, increasing the solubility of compounds in an aqueous medium, was studied using the antiviral drugs mangiferin and favipiravir as an example.

Mangiferin is practically insoluble in water, which, under the gradation of the European pharmacopeia, means the following ratio of substance mass and the solvent volume: more than 10,000 mL per g. The maximum concentration of mangiferin true solution in water that we were able to obtain, proving the absence of micron particles, was only 0.02 mg/mL (Figure 7a, black line). The resulting concentration is lower than the limiting value of the ratio substance mass–solvent volume, given in the pharmacopoeia, which is explained by the existence of polymorphic forms of mangiferin which differ in solubility [60]. Under the same external conditions, but already using an aqueous dilution of the humic complex 1:2000 as a solvent, it was possible to fivefold increase the concentration of mangiferin in the absence of micron particles (Table 1). The method of dynamic light scattering made it possible to detect stable (*n* = 3) particles d = 50 nm in a solution of the substance (0.1 mg/mL) in HC, while maintaining the maximum intensity of particles of 250–300 nm, inherent in HC at a dilution of 1:2000 (Figure 7a, pink line).

Favipiravir is not as sparingly soluble in water as mangiferin, so there was no increase in concentration when the solvent was changed (Table 1). However, dispersion analysis showed a remarkable result: almost a threefold decrease in particle diameter when replacing an aqueous solution of favipiravir with a favipiravir system in a humic complex (Figure 7b). The approved favipiravir drug regimen for the treatment implies huge loading doses (1600–1800 mg) followed by rather a long total duration of therapy with 600–800 mg twice a day. Additionally, despite the data on its good tolerance, such high dosages cannot but cause concern regarding the development of side effects. Potentially, the use of an HC-based transport system could increase the effectiveness of the drug not only due to better penetration of nanoparticles but also through a combined antiviral effect due to the intrinsic activity of HC. In addition, the favipiravir molecule contains in its structure two amide bonds, which, undergoing hydrolysis in an aqueous medium, cause a low pH value of 3.26. When dissolving favipiravir in the humic complex, it is possible to slightly increase this value to 3.34 units.

### 3.4. Second-Order Scattering Quenching as a Method for Evaluating the Process of Incorporation of Mangiferin and Favipiravir into the HC System

As shown earlier (Figure 4), the nanoparticles released upon dilution of the humic complex give rise to strong SOS signals at 2λ_ex_ = λ_em_; this is a characteristic feature of the humic substance. It is of interest to study the sensitivity of this signal to intermolecular interactions when other molecules are included in the supramolecular structures of HC. To further assess the binding affinity between the HC and antiviral drugs, an SOS quenching method was applied.

HC was used as a solvent, its dilution was fixed at 1:2000, corresponding to the dry matter content 3.67 × 10^−^^3^ g/mL, while the drug concentration was varied from 0.02 to 0.18 mmol/L for mangiferin and from 0.03 to 0.32 mmol/L for favipiravir. The scattering intensity of blank HC solution (I_0_) and solutions of tested substances in HC (I) were measured at λ_ex_/λ_em_ = 350 nm/700 nm for mangiferin and λ_ex_/λ_em_ = 365 nm/730 nm for favipiravir. The concentration of drugs was quantified via the peak height (relative scattering intensity), which was obtained by subtracting the blank solution scattering intensity from that of the sample solution. We observed that complexation with mangiferin or favipiravir quenched the SOS signal of the HC at 2 λ_ex_/λ_em_ (Figure 8).

The dependence ΔI–C, mmol/l for both compounds is exponential. With a further increase in the concentration of mangiferin or favipiravir to the maximum values established above, complete quenching of SOS was observed, indicating the effective formation of a stable system of the humic-complex-included drug. The decrease in the SOS signal of the solvent is probably associated with the incorporation of hydrophobic molecules of mangiferin and favipiravir into micelle-like HC fragments and the subsequent decrease in the scattering signal due to the inversion of scattering hydrophobic tails [61].

The proposed SOS method was studied for linearity to subsequently apply this approach to quantify the content of the active substance in a complex matrix of a humate-based delivery system. The decrease in the SOS signal as a function of added substances was examined using logarithmically linearized HC scattering intensity value (Figure 9).

It is shown that the scattering signal, expressed as -logI, linearly depends on the concentration of dissolved mangiferin in the studied concentration range 0–0.18 mmol/L with correlation coefficient of 0.997. For favipiravir, the analytical range 0–0.3 mmol/L was found to be inappropriate. For the values to fall on a linear dependence, it was necessary to reduce the concentration range to a narrower 0.03–0.16 mmol/L (Figure 9b), in this case, the correlation coefficient was 0.989 (Figure 9b, insert).

## 4. Conclusions

Natural HAs have variable compositions, depending on the source and extraction process. Their molecular structure contains numerous groups including carboxylic acids, phenol, enol, alcohol, quinone, ether, and others, which grant them multiple functionalities. The method of fluorescence spectroscopy, when combined with the results of dynamic laser light scattering, made it possible to prove once again that the properties of humic acids are determined by the level of organization of the system. It was also shown that changes in fluorescence, particle size, and the scattering of aqueous solutions of HC were characterized by concentration dependencies with the presence of qualitative rearrangements in the system. Finally, we have shown that humic molecules released from the large supramolecular associations can influence dispersion composition by incorporating some drugs with low water solubility into the nanoparticles. These results point out the feasibility of using HA nanoparticles in complex antiviral therapy as a lipophilic and pH-responsive drug carrier with proven intrinsic antiviral activity. The phenomenon of second-order scattering quenching, which is specifically inherent in aqueous dilutions of the humic complex, can be used to quantify the degree of incorporation of antiviral drugs of various chemical natures into the composition of the carrier complex. The limitations of the method are related to the fluorescence and scattering features of the delivery system based on humins. At low HC dilutions (dry residue content more than 3.67 × 10^−3^ g/mL), we did not observe any significant SOS signal in the HC fluorescence spectrum. As for the tested substances of mangiferin and favipiravir, the method is limited to their rather low concentrations (up to 0.2 mmol/L). At concentrations exceeding this threshold, a complete suppression of the HC scattering signal is observed.

## Figures and Tables

**Figure 1 pharmaceutics-14-00767-f001:**
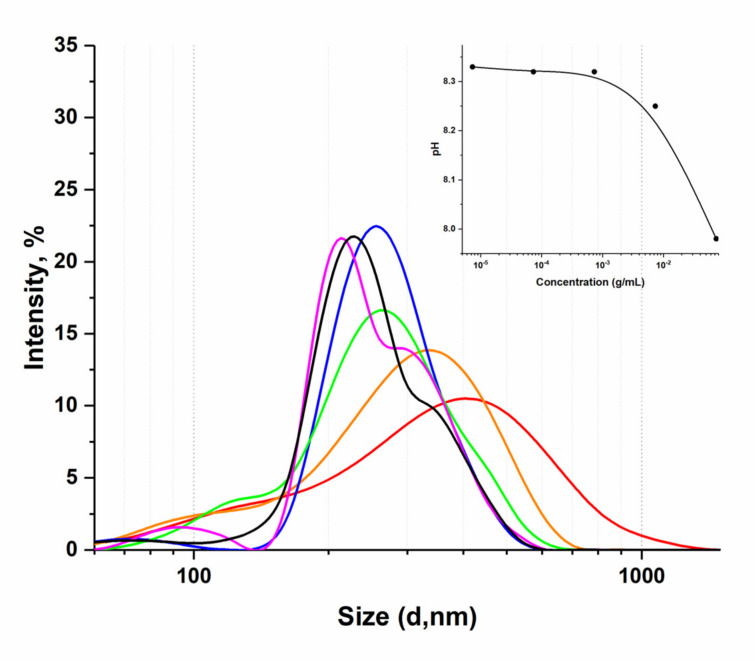
Particle size distribution in nano dispersion of HC dilutions (*v*/*v*): red—1:500 (14.7 × 10^−3^ g/mL); orange—1:1000 (7.34 × 10^−3^ g/mL); blue—1:1500; green—1:2000 (3.67 × 10^−3^ g/mL); black—1:2500 (2.94 × 10^−3^ g/mL); purple—1:3000 (2.45 × 10^−3^ g/mL); pH dependance on concentration (g/mL) is shown in the insert.

**Figure 2 pharmaceutics-14-00767-f002:**
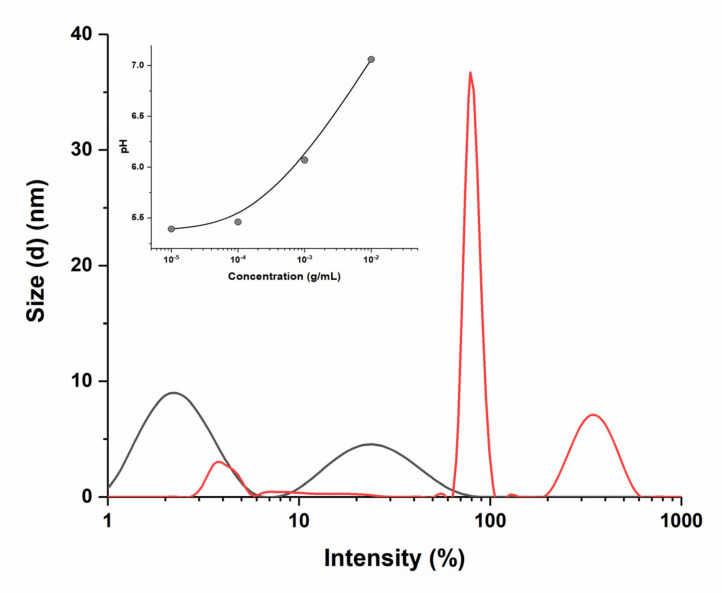
Particle size distribution in BSA colloid system depending on protein concentration (g/mL): red line—1 × 10^−5^ (0.15 μM), black line—1 × 10^−2^ (150 μM); pH dependance on concentration (g/mL) is shown in the insert.

**Figure 3 pharmaceutics-14-00767-f003:**
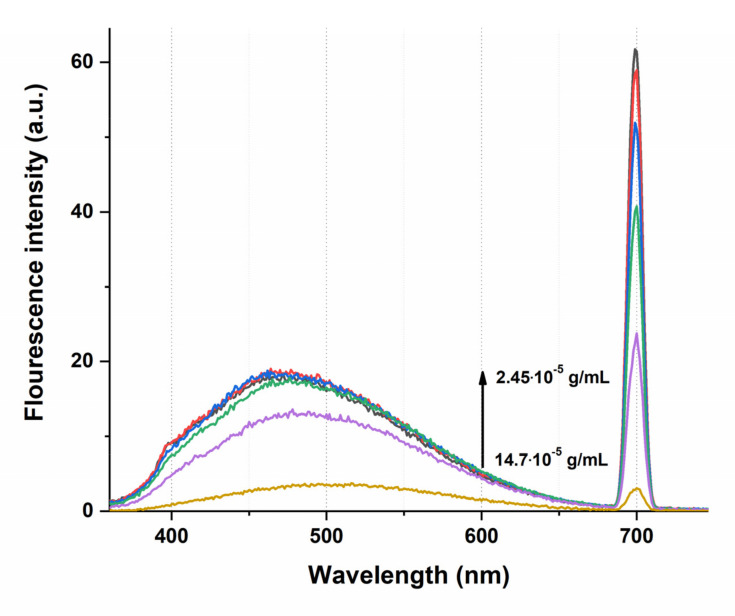
Fluorescence spectrum of HC aqueous dilutions in the concentration range from 14.7 × 10^−^^5^ to 2.45 × 10^−5^ g/mL (1:500–1:3000), λ_ex_ = 350 nm.

**Figure 4 pharmaceutics-14-00767-f004:**
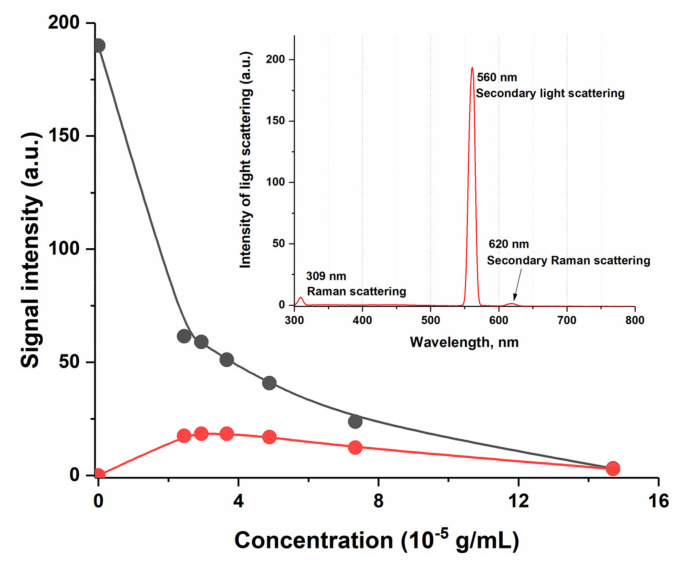
Dependence of the fluorescence signal at 460 nm (red line) and the second-order scattering signal at 700 nm (black line) depending on the concentration of the HC in an aqueous solution, λ_ex_ = 350 nm; the insert shows fluorescence/scattering spectrum of the HC with concentration 7.3 × 10^−14^ g/mL, λ_ex_ = 280.

**Figure 5 pharmaceutics-14-00767-f005:**
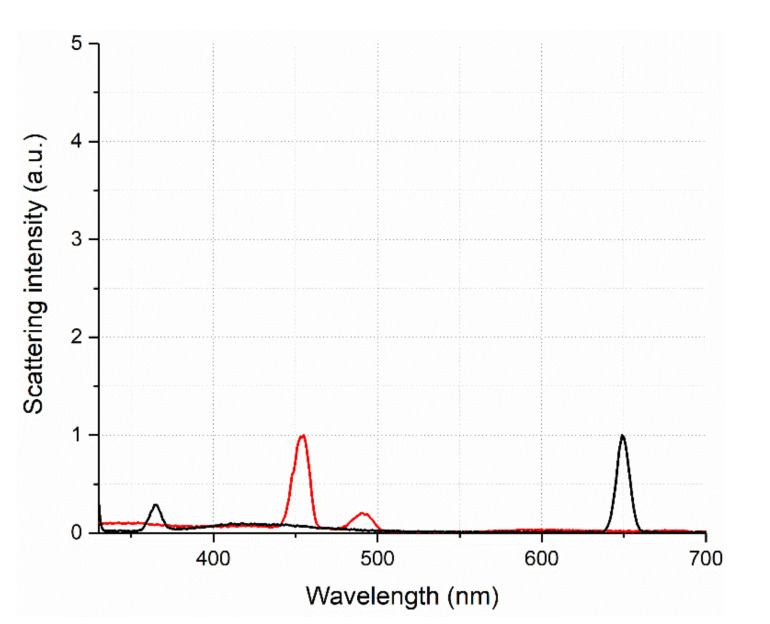
Water scattering signal, red—λ_ex_ = 225 nm, black—λ_ex_ = 325 nm.

**Figure 6 pharmaceutics-14-00767-f006:**
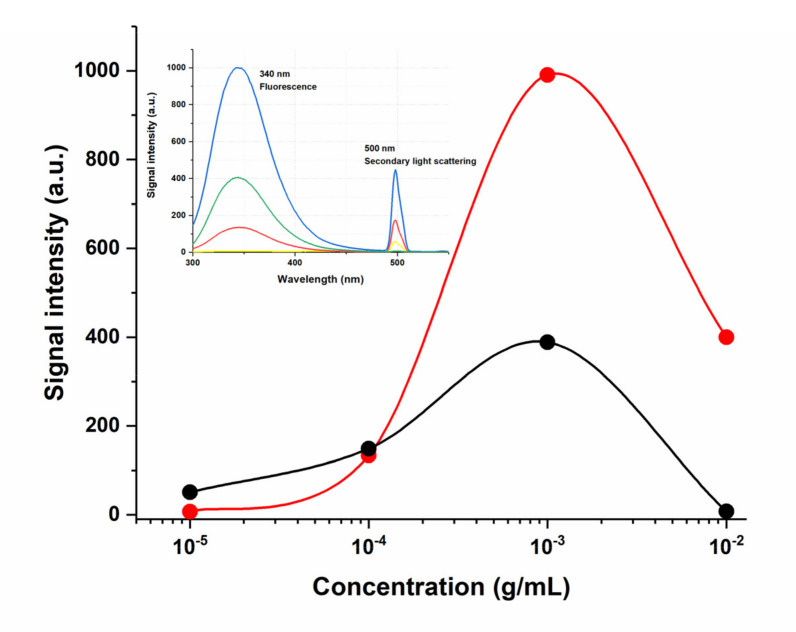
Dependence of the fluorescence signal at 340 nm (red line) and the second-order scattering signal at 500 nm (black line) depending on the concentration of BSA in an aqueous solution, λ_ex_ = 250 nm; the fluorescence spectra of the BSA dilutions are shown in the insert: yellow–10^−5^, red–10^−4^, blue–10^−3^, green–10^−2^, g/mL, λ_ex_ = 250.

**Figure 7 pharmaceutics-14-00767-f007:**
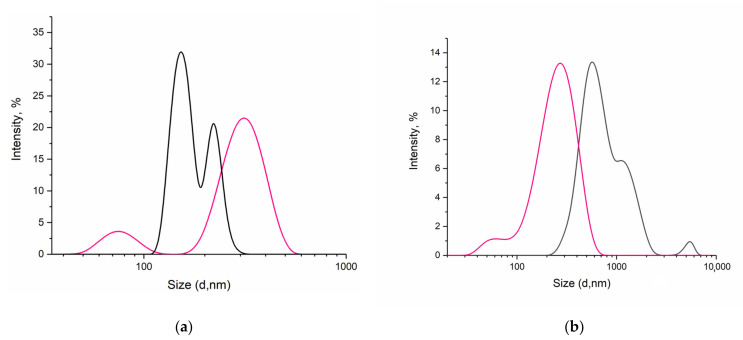
Particle size distribution in mangiferin (**a**) and favipiravir (**b**) solutions: black line—in water, pink line—in HC 1:2000.

**Figure 8 pharmaceutics-14-00767-f008:**
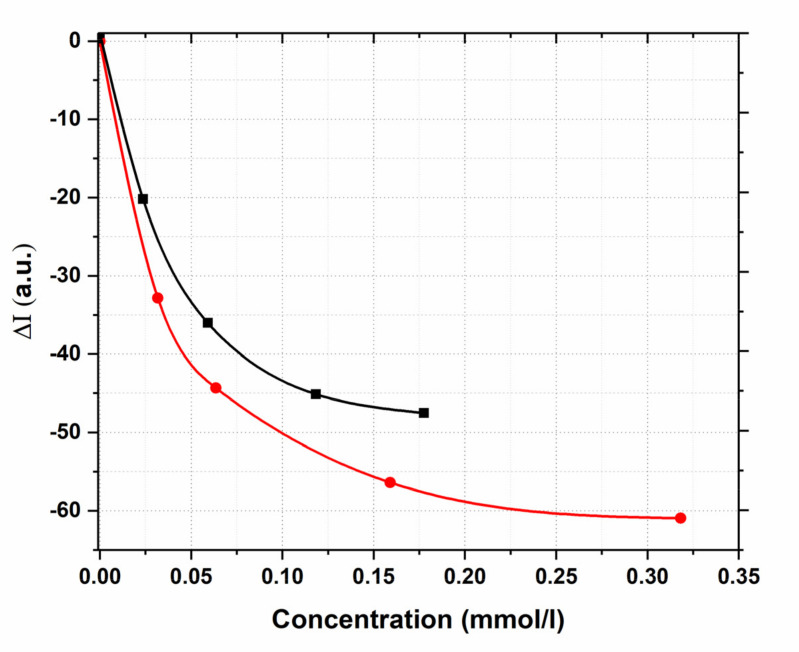
SOS quenching of the humic complex with increasing concentrations of dissolved mangiferin (black line) and favipiravir (red line).

**Figure 9 pharmaceutics-14-00767-f009:**
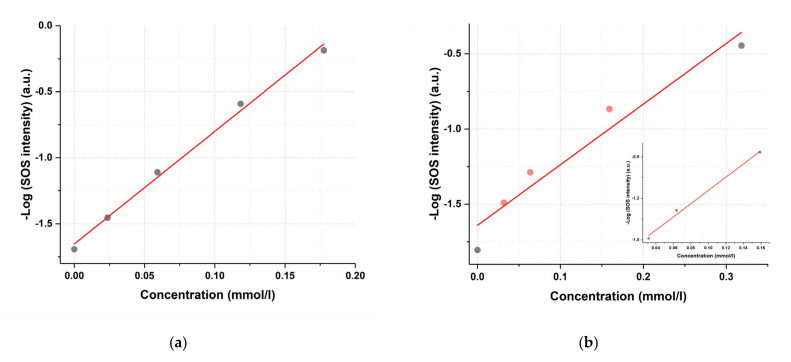
Linearized dependences of the humic complex second-order scattering signal on the concentration of the dissolved preparation: (**a**) for mangiferin (R = 0.997); (**b**) for favipiravir (R = 0.972). The insert in Figure 9b shows the analytical area of the linear dependence for favipiravir–0.03–0.16 mmol/L.

**Table 1 pharmaceutics-14-00767-t001:** Maximum concentrations of mangiferin and favipiravir solutions obtained using different solvents.

Substance	Maximum Concentration			
	In Water		In HC (1:2000)	
	mg/mL	mmol/L	mg/mL	mmol/L
Mangiferin	0.02	0.05	0.1	0.64
Favipiravir	2	4.74	2	4.74

## Data Availability

Not aplicable.

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
