# Peer review of "Second-Order Scattering Quenching in Fluorescence Spectra of Natural Humates as a Tracer of Formation Stable Supramolecular System for the Delivery of Poorly Soluble Antiviral Drugs on the Example of Mangiferin and Favipiravir"

_pharmaceutics, 2022, doi:10.3390/pharmaceutics14040767_

Round 1

Reviewer 1 Report

The study of Morozova et al. describes the development and application of a drug delivery system that was based on the use of humic acid. The manuscript is well written and presents results technically correct. My main concern is regarding the fit of the Stern-Volmer plot (Figure 9). The plot has only three points determining the fitting. Apparently, the curve is not linear and it is important to add more experimental points. It is important to note that the adjustment is not good for these points. 

Author Response

Dear reviewer, we fully agree with your criticism of Figures 9a and 9b.

Since the aim of this work was not to determine the fluorescence quenching constant, we decided to turn away from the Stern–Volmer coordinates and, in order to linearize the dependence, take the scattering intensity values ​​logarithmically. As a result, a linear dependence was obtained for mangiferin in the studied concentration range. For favipiravir, the result was linear in a narrower analytical region. Corrected drawings and conclusion in the file.

Reviewer 2 Report

This paper describes a combination of the method of dynamic light scattering, and fluorescence spectroscopy for the study of the optical properties of aqueous dilutions of the humic substances complex as a potential drug delivery system.

This is a very intersting manuscript for drug delivery applications. It is well-written and organized. I recommend the acceptance of the manuscript after minor revision:

My comments are:

1. Lines 180-185: Add references.

2. Lines 217-220: Add references.

3. Indentify the limitations of this method. 

4. Is this method useful for the characterization of inorganic nanoparticles?

Author Response

1) References were added - 34 and 35

2) Reference was added - 43

3) The limitations are primarily related to the properties of the transport system itself - a significant scattering signal appears only at large dilutions. In addition, we cannot dissolve a large amount of substance in this complex, because when the threshold of 0.2 mmol/L is exceeded, complete suppression of scattering is observed. The described limitations are indicated in the conclusion of the article.

4) The study of inorganic nanoparticles was not the purpose of this work, but we admit such a possibility. In a previous work, where the physical and chemical properties of the humic complex were studied, the presence of bound iron was shown by XRF and IR methods (https://doi.org/10.3390/pharmaceutics13111954). Thus, the possibility of using the humic complex to increase the bioavailability of certain elements due to the formation of nanoparticles is not excluded.
